# Automatic Meniscus Segmentation Using Adversarial Learning-Based Segmentation Network with Object-Aware Map in Knee MR Images

**DOI:** 10.3390/diagnostics11091612

**Published:** 2021-09-03

**Authors:** Uju Jeon, Hyeonjin Kim, Helen Hong, Joonho Wang

**Affiliations:** 1Department of Software Convergence, Seoul Women’s University, Seoul 01797, Korea; ujujeon@swu.ac.kr (U.J.); hyunjinkim@swu.ac.kr (H.K.); 2Department of Orthopedic Surgery, Samsung Medical Center, Sungkyunkwan University School of Medicine, Seoul 06351, Korea; mdwang88@gmail.com

**Keywords:** knee MR images, meniscus segmentation, deep convolutional neural network, adversarial learning, conditional generative adversarial network

## Abstract

Meniscus segmentation from knee MR images is an essential step when analyzing the length, width, height, cross-sectional area, surface area for meniscus allograft transplantation using a 3D reconstruction model based on the patient’s normal meniscus. In this paper, we propose a two-stage DCNN that combines a 2D U-Net-based meniscus localization network with a conditional generative adversarial network-based segmentation network using an object-aware map. First, the 2D U-Net segments knee MR images into six classes including bone and cartilage with whole MR images at a resolution of 512 × 512 to localize the medial and lateral meniscus. Second, adversarial learning with a generator based on the 2D U-Net and a discriminator based on the 2D DCNN using an object-aware map segments the meniscus into localized regions-of-interest with a resolution of 64 × 64. The average Dice similarity coefficient of the meniscus was 85.18% at the medial meniscus and 84.33% at the lateral meniscus; these values were 10.79%p and 1.14%p, and 7.78%p and 1.12%p higher than the segmentation method without adversarial learning and without the use of an object-aware map with the Dice similarity coefficient at the medial meniscus and lateral meniscus, respectively. The proposed automatic meniscus localization through multi-class can prevent the class imbalance problem by focusing on local regions. The proposed adversarial learning using an object-aware map can prevent under-segmentation by repeatedly judging and improving the segmentation results, and over-segmentation by considering information only from the meniscus regions. Our method can be used to identify and analyze the shape of the meniscus for allograft transplantation using a 3D reconstruction model of the patient’s unruptured meniscus.

## 1. Introduction

The menisci is a thin, semi lunar-like tissue pad consisting of the medial meniscus located on the inside of the knee and the lateral meniscus located on the outside of the knee, distributing the load while also reducing friction in the knee [1,2]. Automatic meniscus segmentation from knee MR images is an essential step when analyzing certain aspects of the shape of the meniscus such as the length, width, height, cross-sectional area, and surface area, as required for meniscus allograft transplantation using a 3D reconstruction model of the patient’s unruptured meniscus generated through automatic segmentation of the meniscus. In meniscus allograft transplantation, the most appropriate 3D reconstruction model for the patient can be generated by mirroring the opposite side of the knee of degeneration. However, as shown in Figure 1, automatic meniscus segmentation is challenging due to its thin shape, similar intensity to nearby structures such as the cruciate and collateral ligaments in knee MR images, the considerable shape and size variations of the anterior horn and posterior horn of the meniscus between patients, and due to inhomogeneous intensity levels within the meniscus.

As shown in Table 1, conventional approaches to meniscus segmentation for detecting meniscus damage and for an early diagnosis of osteoarthritis (OA) in knee MR images are divided into intensity-based, shape-based, machine-learning-based, and deep-learning-based methods. When using an intensity-based method, the meniscus is segmented using water-excited double-echo steady-state (weDESS) MR images from the Osteoarthritis Initiative (OAI) database, applying thresholding, morphological operation, and canny edge detection [3]. However, intensity-based methods are associated with the over-segmentation limitation due to leakage into adjacent structures such as bones, cruciate ligaments, and collateral ligaments, which have intensity levels similar to that of the meniscus. With shape-based methods, the active shape model (ASM) is applied to measure similar meniscus for allograft surgery and to diagnose OA [1,4,5]. However, the segmentation accuracy is degraded in shape-based methods in cases involving large shape variations or unusual meniscus shapes.

In machine-learning-based methods, various classifiers including extreme learning machine (ELM), discriminative random field (DRF), k-nearest neighbor (k-NN) classifier, random forest (RF), and patch-based edge classification were applied to segment the meniscus using feature vectors such as intensity values, the eigenvalues of Hessian images, Euclidean distance from the closest bone surface, location information, gradient values of pixels, and texture pattern [6,7,8,9]. However, machine-learning-based methods are also associated with degraded segmentation accuracy when the meniscus has an unusual shape, and they use intensity values and location information as feature vectors.

In deep-learning-based methods, U-Net-based deep learning networks have been applied in most studies. 2D U-Net, and 3D U-Net with various other methods such as auxiliary classifier, residual link, statistical shape model (SSM), 3D conditional random field (CRF), 3D simplex deformable modeling, and attention module have been proposed to segment cartilage and meniscus to assess OA progression [10,11,12,13,14]. However, these deep-learning-based methods did not take into account the class imbalance problem between the meniscus and other structures apart from the meniscus.

In this paper, we propose an automatic meniscus segmentation method that integrates multi-class segmentation networks to localize the meniscus and adversarial-learning-based segmentation networks with an object-aware map to segment the meniscus. The meniscus localization step of the proposed method segments knee MR images into six classes including bone and cartilage to localize the region of the medial and lateral meniscus automatically and to prevent the class imbalance problem between the meniscus and non-meniscus regions. The meniscus segmentation step of the proposed method considers the uniqueness of the shape and intensity distributions of the meniscus for each patient through adversarial learning between a discriminator that judges the segmentation results of a generator and a generator that generates segmented probability maps to prevent the discriminator from making correct decisions in localized regions. In addition, the object-aware map of the proposed method prevents leakage to the surrounding ligaments with intensity levels similar to that of the meniscus by multiplying the original images by corresponding segmented probability maps or ground-truth values and feeding them into the discriminator.

## 2. Materials and Methods

The pipeline of the proposed automatic meniscus segmentation method is illustrated in Figure 2. First, knee structures including bone, cartilage, and meniscus are segmented using whole MR images. Then, the medial meniscus and lateral meniscus are segmented, and the results are improved repeatedly using a conditional generative adversarial network (cGAN) with an object-aware map in the localized meniscus regions-of interest (ROIs).

### 2.1. Materials

A total of 105 normal 3D PD VISTA coronal knee MR images of the opposite side of the knee with reconstructive surgery due to knee joints degeneration acquired from an Achieva 3.0T Philips Medical System were used in this study. This study was approved by the Institutional Review Board of Samsung Medical Center, Seoul, Korea (IRB number: 2010-08-116, 2013-07-097). Our method was evaluated on 26,250 slices from 105 patients aged from 20 to 60 consisting of 89 males and 16 females. The images were of 60 right knees and 45 left knees and were randomly split into 70 patients (17,500 slices) for training, 15 patients (3750 slices) for validation, and 20 patients (5000 slices) for testing. All MR images were obtained from 2011 to 2016 and all had an image resolution of 512 × 512, a pixel size of 0.3125 mm, a slice thickness of 0.5 mm, a repetition time (TR) of 1600 ms, an echo time (TE) of 32.69 ms, and a slice number in the range of 230~250 with the signal intensity normalized through z-score normalization.

### 2.2. Meniscus Localization through Multi-Class Segmentation

Under-segmentation of the meniscus occurs due to the class imbalance problem stemming from the size difference between the meniscus and non-meniscus regions when only the meniscus is segmented in whole-knee MR images due to the thin shape and small size of the meniscus. Thus, knee structures including bone, cartilage, and meniscus were initially segmented to efficiently localize and automatically extract the ROIs of the meniscus considering the anatomical fact that the meniscus is located at the bottom of the femur and femoral cartilage and at the top of the tibia and tibial cartilage.

First, 2D U-Net [15] serves to segment knee MR images into six classes including femoral and tibial bones and cartilage, the meniscus, and the background using whole MR images with a resolution of 512 × 512, as shown in Figure 3d. Here, the contracting path consists of four convolutional layers with two 3 × 3 convolution blocks, batch normalization (BN), a rectified linear unit (ReLU), and 2 × 2 max-pooling. The expanding path consists of four convolutional layers with 2 × 2 up-convolution, two 3 × 3 convolution blocks, BN, and a ReLU with a skip connection between the contracting path and the expanding path. Finally, a 1 × 1 convolution block is added to output segmented probability maps and segmentation results identical in size to the original image.

Second, the ROIs of the medial and lateral meniscus are extracted separately using the meniscus region segmented by 2D U-Net with the margin value of the standard deviation calculated using the minimum and maximum values on the *x*, *y*, and *z* axis for each slice of the meniscus, as shown in Figure 3a–c.

### 2.3. Meniscus Segmentation and Improvement through Adversarial Learning with an Object-Aware Map

Under-segmentation of the meniscus occurs due to inhomogeneous intensity levels within the meniscus, while over-segmentation of the meniscus occurs due to leakage to surrounding structures as the meniscus is adjacent to the femur, tibia, and the cruciate and collateral ligaments. Thus, to segment the medial and lateral meniscus while preventing under- and over-segmentation, adversarial learning using an object-aware map was performed on the localized meniscus ROIs.

First, a generator using 2D U-Net [15] was used to segment the meniscus within the ROIs converted to a resolution of 64 × 64. The images with two classes including the meniscus and background were used as input of a generator due to the prevention of the class imbalance problem between the meniscus and non-meniscus regions by the automatic meniscus localization through multi-class segmentation.

Second, a discriminator consisting of the structures identical to those of the encoder of 2D U-Net [15] with an output layer was utilized to judge and improve the results of the generator. The discriminator consists of five convolutional layers with two 3 × 3 convolution blocks, BN, a ReLU, and max-pooling, while the output layer consists of a 1 × 1 convolution block and a global average pooling step [16]. The loss function used for adversarial learning can be formulated as Equation (1). The loss function consists of adversarial loss as Equation (2) and segmentation loss as Equation (3) based on binary cross-entropy. The adversarial loss in Equation (2) has a large value if the adversarial network can discriminate the output of the generator from the ground truth. The segmentation loss in Equation (3) facilitates the prediction by the generator of the correct class label at each pixel location [17].
(1)$$Loss\;function=\underset{G}{\min}[\max_{D}\;\lambda L_{GAN}\left(G,D\right)+\;L_{SEG}\left(G\right)]$$
(2)$$L_{GAN}=\underset{G}{\min}[\underset{D}{\max}\left[logD\left(x,y\right)\right]+\left[\log\left(1-D\left(x,G\left(x\right)\right)\right)\right]$$
(3)$$L_{SEG}=\underset{G}{\min}\left[y\cdot logG\left(x\right)-\left(1-y\right)\cdot log\left(1-G\left(x\right)\right)\right]$$

Here, G is the generator network; D is the discriminator network; $L_{GAN}$ is the adversarial loss; $L_{SEG}$ is the segmentation loss; x is the original image; y is the ground truth; and λ is a hyper-parameter for good balance between the adversarial loss and the segmentation loss. We used D(x,y) and D(x,G(x)) to denote the probability estimate in the range of 0 to 1 regarding whether y or G(x) comes from the ground truth or the segmented probability map generated by the generator, and G(x) to denote the segmented probability map produced by the generator. In the experiments, λ was set to 0.04.

With reference to the adversarial loss, the generator was trained to make the discriminator determine the segmented probability maps as the ground truth, and the discriminator was trained to maximize the probability of assigning correct labels to both the training examples and the samples from the generator [18]. Regarding the segmentation loss, the generator was trained to minimize the differences in the probability distributions between the segmented probability maps and the ground truth. As shown in Figure 4b, object-aware maps, for which the original images were multiplied by the corresponding segmented probability maps or the ground truth, were fed into the discriminator to focus more on the meniscus region by considering only information pertaining to the meniscus and not useless background information. In conventional adversarial networks, the original images are concatenated with the corresponding segmented probability maps or the ground truth, as shown in Figure 4a.

The proposed adversarial learning approach can prevent under-segmentation due to the inhomogeneity of intensity levels by segmenting the meniscus in localized ROIs and repeatedly judging and improving the segmentation results. The proposed object-aware map can focus more on meniscus regions and prevent leakages to nearby organs with intensity levels similar to that of the meniscus by multiplying the original images by the corresponding probability maps or the ground truth and feeding these outcomes as input into the discriminator with information only from the meniscus regions.

## 3. Results

In Figure 5, the first and second columns show the results of adversarial learning for the medial and lateral meniscus, respectively. It can be observed that under-segmentation results due to inhomogeneous intensity levels are prevented through adversarial learning by repeatedly judging and improving the segmentation results.

Figure 6 shows the results of a comparison between conventional adversarial learning and adversarial learning with an object-aware map. In Figure 6, the first and second columns show the results for the medial meniscus, and the third and fourth columns show the results for the lateral meniscus. As shown in Figure 6b, an outlier occurs in the posterior and anterior cruciate ligament sections and over-segmentation arises with the collateral ligament and meniscus boundary because their intensity levels are similar to that of the meniscus in the conventional adversarial network. It can be observed that outliers and over-segmentation results are prevented when using an object-aware map, as shown in Figure 6c.

To evaluate the meniscus segmentation performances, we compared the proposed method (Method D) with segmentation results using 2D U-Net with two classes including the meniscus and background in the whole MR image (Method A), segmentation results using 2D U-Net within automatically localized meniscus ROIs (Method B) and segmentation results using a conventional adversarial network within automatically localized meniscus ROIs (Method C). For a qualitative evaluation, the results of meniscus segmentation by applying the comparison methods and the proposed method were compared with the ground truth. For a quantitative evaluation, the Dice similarity coefficient (DSC), accuracy, sensitivity, specificity, positive predictive value (PPV), and negative predictive value (NPV) were used to measure the gap between the ground truth and the segmented probability maps used in the comparison methods and the proposed method, as shown in Equation (4).
(4)$$DSC=\frac{2TP}{2TP+FN+FP}\times 100\;Accuracy=\frac{TP+TN}{TP+FP+TN+FN}\times 100\;Sensitivity=\frac{TP}{TP+FN}\times 100\;Specificity=\frac{TN}{TN+FP}\times 100\;PPV=\frac{TP}{TP+FP}\times 100\;NPV=\frac{TN}{TN+FN}\times 100$$

In this equation, TP (true positive) denotes the number of pixels correctly predicted as meniscus pixels, TN (true negative) is the number of pixels correctly predicted as non-meniscus pixels, FP (false positive) is the number of pixels incorrectly predicted as meniscus pixels, and FN (false negative) represents the number of pixels incorrectly predicted as non-meniscus pixels.

Figure 7 shows the qualitative performance of Method A, Method B, Method C, and Method D. Method A showed under-segmentation due to the thin shape and small size of the meniscus. Method B prevented under-segmentation by segmenting knee structures into six classes, but showed an outlier due to leakage to the anterior and posterior cruciate ligaments and the background with intensity similar to that of the meniscus also showed under-segmentation results due to inhomogeneous intensity areas within the meniscus. Method C prevented under-segmentation compared to Method B by repeatedly judging and improving the segmentation results through adversarial learning, but it still showed over-segmentation due to leakage to adjacent organs with areas of similar intensity to the meniscus. Method D prevented under- and over-segmentation through adversarial learning using an object-aware map.

Table 2 shows the quantitative evaluation results of meniscus segmentation. As shown in the table, the results of the sensitivity and NPV of Method B were correspondingly higher by 0.87 percentage points (%p) and 0.45%p at the medial meniscus and by 2.66%p and 0.38%p at the lateral meniscus compared to those of Method A due to the prevention of under-segmentation through the segmentation of knee structures in the whole MR image. The results of the sensitivity and NPV of Method C were correspondingly higher by 0.64%p and 0.03%p at the medial meniscus and by 1.55%p and 0.08%p at the lateral meniscus compared to those of Method B due to the prevention of under-segmentation through adversarial learning. The DSC, PPV, specificity, and accuracy for Method D were correspondingly higher by 1.12%p, 2.75%p, 0.24%p, and 0.16%p for the medial meniscus and by 1.14%p, 2.70%p, 0.38%p, and 0.20%p for the lateral meniscus compared to Method C because Method D prevented over-segmentation through adversarial learning using an object-aware map. However, the sensitivity and NPV of Method D showed lower results by 1.83%p and 0.08%p for the medial meniscus and by 3.73%p and 0.19%p for the lateral meniscus compared to Method C due to under-segmentation stemming from the object-aware map by which the discriminator determined the segmentation performance with information only about the meniscus without considering the background information. The sensitivity and NPV of Method C showed the highest results given that over-segmentation occurs in a nearby organ with an intensity level similar to that of the meniscus. The performance difference between medial and lateral meniscus segmentation by Method C was significantly improved from 3.86%p to 0.87%p compared to Method B, and Method D showed the smallest performance difference of 0.85%p. When using Method C, the standard deviations of the DSC, PPV, specificity, and accuracy were more stabilized than those of Method B, and Method D had the lowest standard deviation values for the DSC, PPV, specificity, and accuracy.

## 4. Discussion and Conclusions

In this paper, we automatically extracted the ROIs of the meniscus through a multi-class segmentation method for bone and cartilage areas and improved the segmentation performance through adversarial learning using an object-aware map by iterative evaluations of the segmentation results. The automatic meniscus localization through multi-class segmentation can improve segmentation performance and prevent the class imbalance problem by focusing on local regions. The proposed adversarial learning approach can prevent under-segmentation due to the inhomogeneity of intensity levels by segmenting the meniscus in localized ROIs and repeatedly judging and improving the segmentation results. The proposed object-aware map can focus more on meniscus regions and prevent leakages to nearby organs with intensity levels similar to that of the meniscus by multiplying the original images by the corresponding probability maps or the ground truth and feeding these outcomes as input into the discriminator with information only from the meniscus regions. Our meniscus localization method can prevent outliers and can extract reasonable ROIs by segmenting whole MR images into six classes including bone and cartilage instead of only segmenting the meniscus. Our meniscus segmentation method can also prevent under-segmentation due to areas of inhomogeneous intensity within the meniscus given its use of adversarial learning and can prevent over-segmentation due to leakages to adjacent structures with intensity levels similar to that of the meniscus with its use of an object-aware map.

Multiview data have recently been used for considering spatial information, and attention has recently been used for focusing on the target [19,20,21,22]. Yang et al. proposed a multiview learning network to consider spatial correlation between 2D slices and the attention mechanism in the dilated attention network to force the model to focus on the locations of the left atrium (LA) scar for LA scar segmentation [19]. Liu et al. proposed the segmentation algorithm consists of three sub-networks consisting of improved ResNet50 for the encoding of rich semantic information from original images, the feature pyramid attention network to help capture the information at multiple scales, and the decoder network to recover the spatial information for prostate zonal segmentation [20]. Li et al. proposed recurrent aggregation network with multiview images as input that contained a segmentation branch for echocardiographic sequence segmentation and classification branch for classification of views of input data to extract additional information for the representation of the diversity of different views [21]. Wu et al. proposed U-Net-based network with magnitude and velocity data with multi-channel attention block to better incorporate the multi-channel information for myocardium segmentation [22].

Adversarial learning has recently been used for various purposes such as segmentation and domain generation [23,24]. Chen et al. proposed an inter-cascaded generative adversarial network, namely JAS-GAN, to segment the unbalanced atrial targets from late gadolinium-enhanced cardiac magnetic resonance (LGE CMR) images [23]. JAS-GAN consists of an adaptive attention cascade network to correlate the spatial location of LA and small atrial scars and a joint discriminative network to regularize the adaptive attention cascade network to produce a matched joint distribution of unbalanced atrial targets. Chen et al. also proposed a discriminative consistent domain generation (DCDG) approach to achieve semi-supervised learning by fusing the feature spaces of labeled and unlabeled data through double-sided domain adaptation [24]. In our study, we used the adversarial network for meniscus segmentation to train the generator to determine the segmented label as the ground truth by the discriminator while preventing under-segmentation due to inhomogeneous intensity levels within the meniscus and over-segmentation due to leakage to surrounding structures adjacent to the meniscus.

The medical image fusion methods have been used to obtain a fine image by extracting each feature using multi-modality images and enhancing detailed information [25]. Zhu et al. proposed a multi-modality medical image fusion framework based on phase congruency and local Laplacian energy to better extract the details of medical images and save more image energy to enhance detailed features such as texture and edge information while preserving the structured information of source multi-modality images [25]. However, it is difficult to make up for the limitations of the image itself if only single modality is used, even if the image is enhanced through pre-processing. Thus, we tried to compensate for the limitations in the deep learning process by using the original image.

In the experiments, the average DSC was 85.18% for the medial meniscus and was 84.33% for the lateral meniscus, values that were improved by 10.79%p and 7.78%p compared to a method without adversarial learning and by 1.14%p and 1.12%p compared to a method without an object-aware map for the medial and lateral meniscus, respectively. As shown in Figure 7, the under-segmentation still occurred in blurry boundary area with high uncertainty. To address this limitation, the uncertainty estimation applying in the training process will be implemented to improve the segmentation performance in areas with high uncertainty as future work. Our method can be used to accurately segment the normal meniscus without degeneration by identifying and analyzing the shape of the meniscus and to generate the most appropriate meniscus model for allograft transplantation when the meniscus is damaged or missing using a 3D bioprinting reconstruction model of the patient’s unruptured meniscus.

## Figures and Tables

**Figure 1 diagnostics-11-01612-f001:**
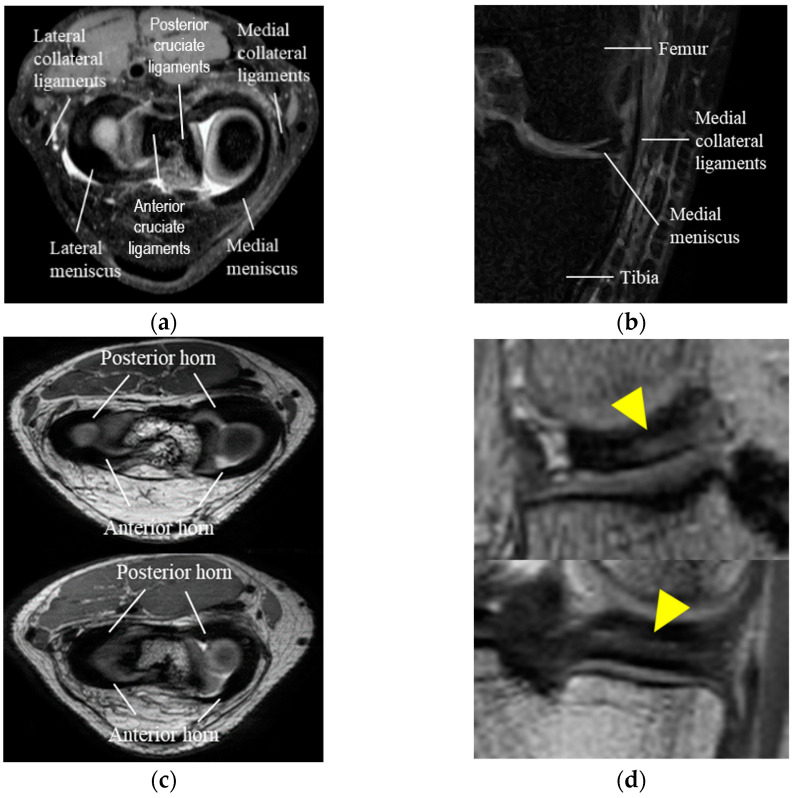
Characteristics of the meniscus in knee MR images: (**a**) Structures of the lateral and medial meniscus in the axial view; (**b**) Similar intensity of the medial meniscus and collateral ligaments in the coronal view; (**c**) Shape variances of anterior and posterior horns of the meniscus among patients; (**d**) Inhomogeneous intensity of lateral and medial meniscus (yellow arrow) in knee MR images.

**Figure 2 diagnostics-11-01612-f002:**
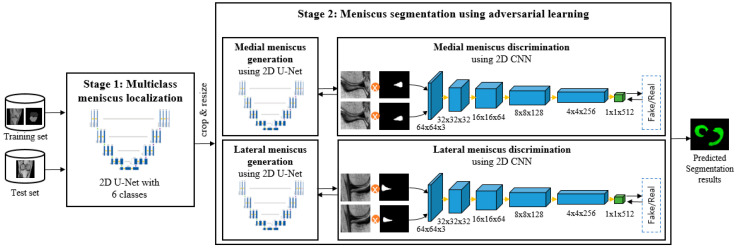
Pipeline of the proposed method.

**Figure 3 diagnostics-11-01612-f003:**
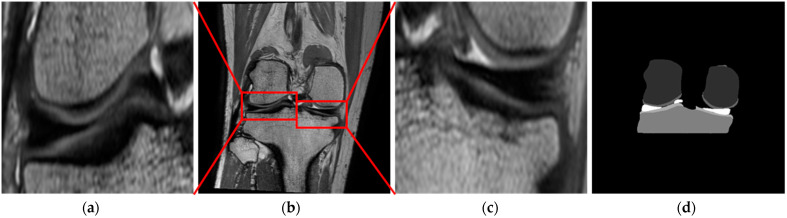
Automatic localization of the meniscus. (**a**) The medial meniscus localization. (**b**) Original image. (**c**) The lateral meniscus localization. (**d**) The 6-classes segmentation results (dark grey: femur, medium grey: femoral cartilage, grey: tibia, light grey: tibial cartilage, white: meniscus).

**Figure 4 diagnostics-11-01612-f004:**
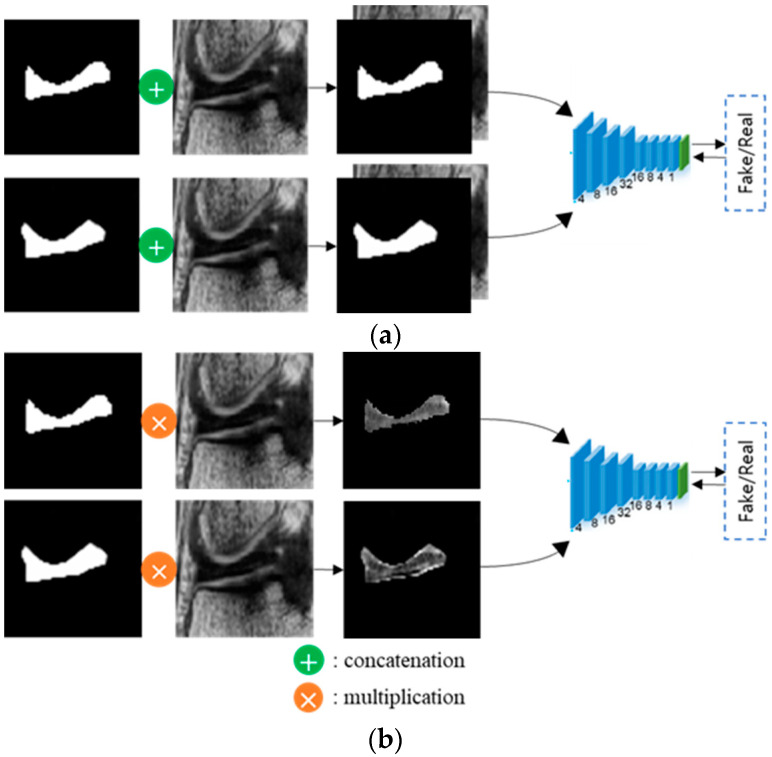
Architecture of the discriminator of the conventional adversarial network and the proposed adversarial network using an object-aware map: (**a**) Conventional adversarial network; (**b**) Adversarial network using object-aware map.

**Figure 5 diagnostics-11-01612-f005:**
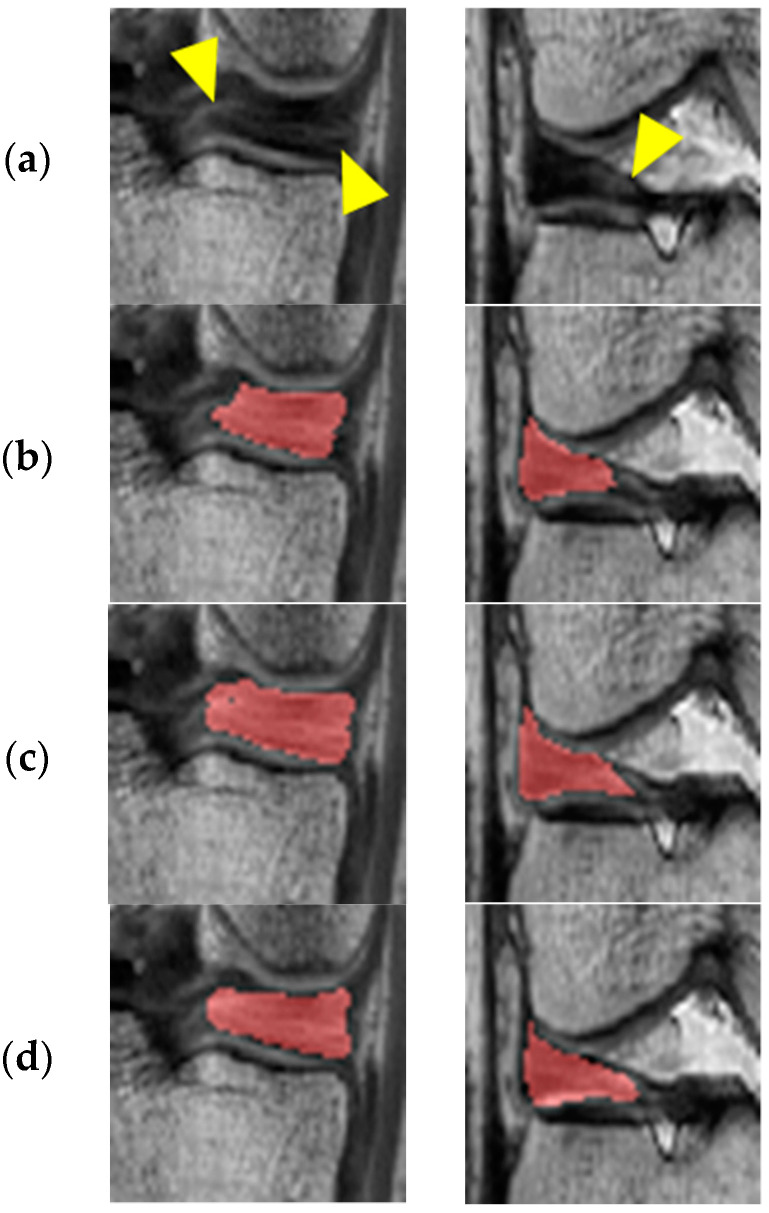
Segmentation results of before and after adversarial learning: (**a**) Original image; (**b**) Before adversarial learning; (**c**) After adversarial learning; (**d**) Ground truth.

**Figure 6 diagnostics-11-01612-f006:**
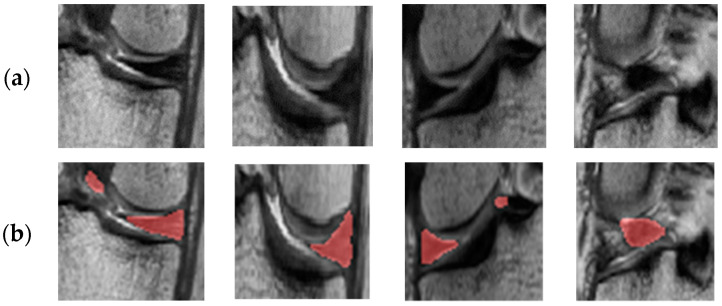
Segmentation results of conventional adversarial network and adversarial network using an object-aware map: (**a**) Original image; (**b**) Conventional adversarial network; (**c**) Adversarial network using object-aware map; (**d**) Ground truth.

**Figure 7 diagnostics-11-01612-f007:**
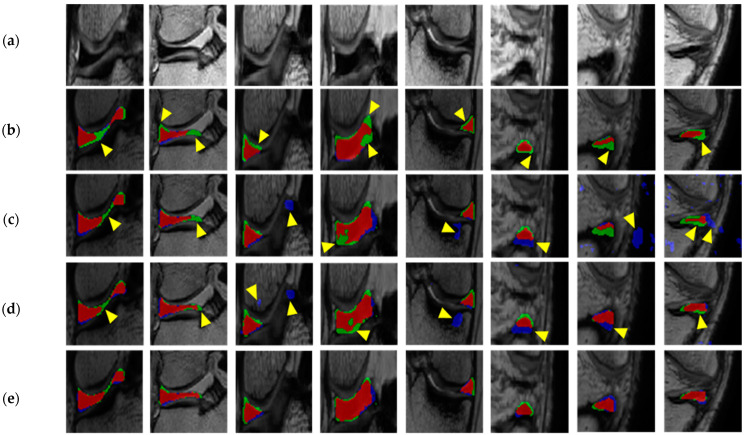
Qualitative evaluation of meniscus segmentation results: (**a**) Test images; (**b**) Results of Method A; (**c**) Results of Method B; (**d**) Results of Method C; (**e**) Results of Method D. (Red: overlay area, Green and Blue: under- and over-segmentation areas, respectively).

**Table 1 diagnostics-11-01612-t001:** Related works of meniscus segmentation.

Approaches	Authors	Dataset (Patients)	Methods	Performance (DSC ^1^, %)
Intensity-based	Swanson et al. [3]	OAI (24)	Thresholding and Morphological operation	LM ^2^: 80.00
Shape-based	Kim et al. [1]	Local (18)	Statistical Shape Model & Active Shape Model & Interpolated shape information	MM ^3^: 0.54 (ADD ^4^, mm)/LM: 0.73 (ADD, mm)
	Fripp et al. [4]	Local (14)	Active Shape Model	MM: 77.00/LM: 75.00
	Paproki et al. [5]	OAI (88)	Active Shape Model	MM: 78.30/LM: 83.90
Machine Learning-based	Zhang et al. [6]	Local (11)	ELM and DRF	LM + MM: 81.96
	Dam et al. [7]	OAI (88)	Voxel-wise k-NN	MM: 76.10/LM: 82.20
	Saygili et al. [8]	OAI (88)	ELM and RF	LM + MM: 82.73
	Kim et al. [9]	Local (10)	Multi-atlas-based locally-weighted voting & Patch-based edge classification	MM: 80.13/LM: 80.81
Deep Learning-based	Norman et al. [10]	OAI (87)	2D U-Net	MM: 73.10/LM: 81.20
	Raj et al. [11]	OAI (88)	3D μ-Net	MM: 80.18/LM: 84.93
	Tack et al. [12]	OAI (88)	2D U-Net and Statistical Shape Model and 3D U-Net	MM: 83.14/LM: 88.25
	Zhou et al. [13]	OAI (20)	2D SegNet and CRF	LM + MM: 83.10
	Byra et al. [14]	Local (61)	2D Attention U-Net	MM: 84.10/LM: 82.90

^1^ DSC: Dice similarity coefficient; ^2^ LM: Lateral meniscus; ^3^ MM: Medial meniscus; ^4^ ADD: Average distance difference.

**Table 2 diagnostics-11-01612-t002:** Quantitative evaluation of medial and lateral meniscus segmentation results (bold indicates the best performance of the comparison methods).

	DSC (%)	PPV (%)	Specificity (%)	Accuracy (%)	Sensitivity (%)	NPV (%)
Medial Meniscus						
Method A	79.99 (±5.14)	78.89 (±6.57)	98.71 (±0.72)	97.95 (±0.59)	87.09 (±3.92)	98.95 (±0.51)
Method B	77.40 (±6.22)	69.69 (±9.36)	98.12 (±0.60)	97.65 (±0.53)	87.96 (±4.04)	99.40 (±0.19)
Method C	84.06 (±3.54)	81.39 (±5.92)	98.92 (±0.45)	98.43 (±0.47)	**88.60 (±4.90** **)**	**99.43 (±0.26** **)**
Method D	**85.18 (±2.74)**	**84.14 (±5.92)**	**99.16 (±0.42)**	**98.59 (±0.35)**	86.77 (±4.89)	99.35 (±0.24)
Lateral Meniscus						
Method A	79.19 (±4.08)	78.28 (±5.62)	98.96 (±0.47)	97.94 (±0.44)	85.14 (±3.51)	98.98 (±0.47)
Method B	73.54 (±6.32)	63.85 (±8.73)	97.57 (±0.59)	97.09 (±0.66)	87.80 (±5.17)	99.36 (±0.36)
Method C	83.19 (±2.05)	80.90 (±6.27)	98.81 (±0.22)	98.32 (±0.33)	**89.35 (±4.12** **)**	**99.44 (±0.27** **)**
Method D	**84.33 (±2.60)**	**83.60 (±4.81)**	**99.19 (±0.20)**	**98.52 (±0.30)**	85.62 (±5.93)	99.25 (±0.41)

## Data Availability

Not applicable.

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
