# Peer review of "Automatic Meniscus Segmentation Using Adversarial Learning-Based Segmentation Network with Object-Aware Map in Knee MR Images"

_diagnostics, 2021, doi:10.3390/diagnostics11091612_

Round 1

Reviewer 1 Report

This paper proposes a automatic meniscus segmentation method using adversarial learning based segmentation network with object-aware map in knee MR images. Specifically, this method uses U-Net to segment the knee joint image into six classes containing background. Then, adversarial learning with an object-aware map for the medial and lateral meniscus can improve the problems of under-segmentation and over-segmentation.

  1. The lines 99 to 100 of the introduction mention "a generator that generates segmented probability maps to prevent the discriminator from making correct decisions in localized regions.". "prevent ... correct decisions?" Is there a problem with this statement?
  2. In the first stage, the knee joint image is segmented into 6 classes, but most of the classes seem to be unused in the second stage. It would be better to explain the intention.
  3. It is better to add comparative experiments with some current advanced methods to show the superiority of the proposed method.
  4. What is the significance of the second stage meniscus and meniscus treatment separately?
  5. It is better to have labels attached to all index calculation formulas.
  6. More recently published solutions should be discussed. For example, Z. Zhu, M. Zheng, G. Qi, D. Wang and Y. Xiang, "A Phase Congruency and Local Laplacian Energy Based Multi-Modality Medical Image Fusion Method in NSCT Domain," in IEEE Access, vol. 7, pp. 20811-20824, 2019, doi: 10.1109/ACCESS.2019.2898111.The authors should compare the proposed method with it carefully. 

Reviewer 2 Report

The authors reported a method using two-stage deep learning that combines a 2D U-Net-based meniscus localization network with a conditional generative adversarial network-based segmentation network using an object-aware map. The paper is well written with interesting results showing. However, there are still some major concerns of this work:

1. The GAN based segmentation method has been proposed in similar LGE MRI data segmentation tasks, for example:

Chen, Jun, et al. "JAS-GAN: Generative Adversarial Network based joint atrium and scar segmentation on unbalanced Atrial targets." IEEE Journal of Biomedical and Health Informatics (2021).

Chen, Jun et al. "Discriminative consistent domain generation for semi-supervised learning." In International Conference on Medical Image Computing and Computer-Assisted Intervention, pp. 595-604. 2019.

The authors may refer to these studies and highlight the novelties of the current study.

2. Other related studies that the authors may need to discuss especially with the lastest studies using attention and multiviews of MRI:

Yang, Guang, et al. "Simultaneous left atrium anatomy and scar segmentations via deep learning in multiview information with attention." Future Generation Computer Systems 107 (2020): 215-228.

Liu, Yongkai, et al. "Automatic prostate zonal segmentation using fully convolutional network with feature pyramid attention." IEEE Access 7 (2019): 163626-163632.

Li, Ming, et al. "MV-RAN: Multiview recurrent aggregation network for echocardiographic sequences segmentation and full cardiac cycle analysis." Computers in biology and medicine 120 (2020): 103728.

Wu Y et al. Fast and Automated Segmentation for the Three-Directional Multi-Slice Cine Myocardial Velocity Mapping. Diagnostics. 2021 Feb 19;11(2):346. doi: 10.3390/diagnostics11020346.

Zhang, Wenbo, et al. "ME‐Net: Multi‐encoder net framework for brain tumor segmentation." International Journal of Imaging Systems and Technology (2021).

3. How about the uncertainty of the segmentation?

Liu, Yongkai, et al. "Exploring Uncertainty Measures in Bayesian Deep Attentive Neural Networks for Prostate Zonal Segmentation." IEEE Access 8 (2020): 151817-151828.

4. What is the novelty of the proposed GAN? Here are some GAN related work that the authors may want to discuss:

Hao, Jingyu, et al. "Annealing genetic GAN for minority oversampling." BMVC (2020).

Lv, Jun, et al. "PIC-GAN: A Parallel Imaging Coupled Generative Adversarial Network for Accelerated Multi-Channel MRI Reconstruction." Diagnostics 11.1 (2021): 61.

Lv, Jun, et al. "Which GAN? A comparative study of generative adversarial network-based fast MRI reconstruction." Philosophical Transactions of the Royal Society A 379.2200 (2021): 20200203.

Yuan, Zhenmou, et al. "SARA-GAN: Self-Attention and Relative Average Discriminator Based Generative Adversarial Networks for Fast Compressed Sensing MRI Reconstruction." Frontiers in Neuroinformatics 14 (2020).

Wang, Chengjia, et al. "DiCyc: GAN-based deformation invariant cross-domain information fusion for medical image synthesis." Information Fusion 67 (2021): 147-160.

Wang, Chengjia, et al. "SaliencyGAN: Deep learning semisupervised salient object detection in the fog of IoT." IEEE Transactions on Industrial Informatics 16.4 (2019): 2667-2676.

5. The authors may want to publish their codes considering the benefit for the community. Besides, it can help with the testing of reproducibility of the proposed method. 

6. Also such clinical related study may suffer problems of reproducibility. The implementation should ideally be tested on an external datasets.

7. The study seems lack of ablation studies and quantitative results.

Reviewer 3 Report

This paper presents a Deep Learning algorithm that automatically segments 6 labels from a 2D knee MRI. The architecture is based on a UNet combined with a GAN.

Major comments:

  1. Have the authors tested the accuracy of any of the other available architectures / approaches with their dataset? If not, then why create another architecture? I recommend testing each label’s segmentation with at least 4 prevalent architectures to also prove the need for a new algorithm.
  2. How does the proposed architecture perform relative to existing architectures using the presented dataset? Using the same dataset with different architectures will allow for a more reliable comparison of performance.

Minor comments:

  1. Lines 72-86: instead of listing one paper after the next, please consolidate and synthesize the information to make a coherent story with a conclusion.
  2. Please define all terms and acronyms before using them: LM, MM, %p …
  3. There are many run-on or unnecessarily long sentences, especially in the introduction, making it challenging to read and follow. Please correct.

Reviewer 4 Report

The aim of the present study is to introduce a new approach for automatic meniscus segmentation by combining 2D U-Net-based meniscus localization network with a conditional generative adversarial network-based segmentation network utilizing an object-aware map. The most important outcome of this study is that this approach can prevent under- and over-segmentation of the menisci.

I strongly recommend to give a deep check on the entire text (spelling, grammar, etc.) by a native speaker.

The methodology is written in a very detailed and understandable way. However, in this section, already some results and discussion points have been described. I strongly recommend to rearrange the manuscript in accordance to the conventional manuscript sections (Material and Methods, Results and Discussion). I therefore recommend to rearrange the manuscript. Because of this structural and some further I cannot recommend your manuscript for publication in mpdi-diagnostics in its current form. Please find the detailed comments / questions below:

Abstract:

Please provide a conclusion at the end of your abstract.

Introduction:

l.30      A meniscus does not consist of a lateral and medial meniscus. Within the knee, there is a lateral and a medial one. Please revise the formulation

l.40      Variations become even worse with increasing OA and meniscus degeneration. Unusual meniscus shapes can also be due to tissue degeneration. Maybe you can add this here!

Maybe you could mention in the first paragraph of the introduction, that automatic segmentation is also the basis for generating patient-specific models.

Materials and Methods:

l.114    Did any of the 105 patients showed signs of meniscal degeneration? Can you please comment on that?

Please check again your Materials and Methods part. In some sections you show and describe results (for example l198-l214) which should be definitely moved to the results part. Further, your also discuss in some parts (for example l151-153; l217-l223) your approach. Please move this to your discussion!

Results:

l.225 – 239 should be moved to the methods part, because you describe the different measurement quantities, which have been used to evaluate the segmentation performance.  

Discussion:

Please discuss your approach in the context of patient specific modeling (or at least mention it in your conclusion); accuray of segmentation in severe degenerated knee joints

Please comment on limitations of your approach

Round 2

Reviewer 3 Report

Accept the revisions 

Reviewer 4 Report

The extensive revision of the manuscript, along with addressing all comments / concerns of the reviewers improved the manuscript significantly.  Congratulations.

I only have one open comment / recommendation regarding Point 2 (manuscript 36-38):

The menisci are definitely NOT originating from ONE tissue pad. The lateral and medial knee joint meniscus are two independent structures, which are slightly connected via the Lig. transversum. My previous comment was not focussing on the differentiation between the lateral and medial meniscus but on the well-known fact that these are TWO INDEPENDENT structures...  I strongly recommend to correct this in your manuscript as it reads as if the two menisci originate from one tissue pad.

You can crosscheck e.g. here:

https://www.ncbi.nlm.nih.gov/pmc/articles/PMC3435920/

 Or, alternatively at any of the references you mentioned in your previous reply.